# Pruning neural networks using FishLeg estimation

## Abstract

In many domains, the most successful AI models tend to be the largest, indeed often too large to be handled by AI players with limited computational resources. To mitigate this, a number of compression methods have been developed, including methods that prune the network down to high sparsity whilst retaining performance. The best-performing pruning techniques are often those that use second-order curvature information (such as an estimate of the Fisher information matrix) to score the importance of each weight and to predict the optimal compensation for weight deletion. However, these methods are difficult to scale to high-dimensional parameter spaces without making heavy approximations. Here, we propose the FishLeg surgeon (FLS), a new second-order pruning method based on the Fisher-Legendre (FishLeg) optimizer. At the heart of FishLeg is a meta-learning approach to amortising the action of the *inverse* FIM, which brings a number of advantages. Firstly, the parameterisation enables the use of flexible tensor factorisation techniques to improve computational and memory efficiency without sacrificing much accuracy, alleviating challenges associated with scalability of most second-order pruning methods. Secondly, directly estimating the inverse FIM leads to less sensitivity to the amplification of stochasticity during inversion, thereby resulting in more precise estimates. Thirdly, our approach also allows for progressive assimilation of the curvature into the parameterization. In the gradual pruning regime, this results in a more efficient estimate refinement as opposed to re-estimation. We revisit the autoencoder optimisation benchmark of the original FishLeg paper and show that FLS yields highly effective one-shot and gradual pruning, better than previous methods. We further extend FishLeg by developing new structured approximations of the inverse Fisher for convolutional layers. We find that FishLeg greatly improves one-shot pruning accuracy over previous second-order methods on ResNet50 (e.g. 62% accuracy at 75% sparsity, v.s. 41% for M-FAC).

## 1 Introduction

The current staggering growth of AI models is threatening to sideline small and medium-sized AI contributors with limited access to compute resources who cannot afford to run the largest models. Consequently, there is a growing need for methods that can compress these models down to a fraction of their original size whilst retaining their performance (Liu & Wang, 2023).

Here, we focus on unstructured network pruning, i.e. the process of zeroing out as many weights as possible without substantially impacting the quality of the model. We build on the Optimal Brain Surgeon (OBS; LeCun et al., 1989; Hassibi & Stork, 1992), a classical approach to pruning that approximates the network's loss function in quadratic form to determine (i) the importance of each weight and (ii) the optimal way of compensating for their deletion. Several recent studies have shown that second-order importance scores are more accurate than scores derived from weight magnitudes and/or gradients (Gale et al., 2019; Sanh et al., 2020), yielding more effective pruning in convolutional (Theis et al., 2018; Singh & Alistarh, 2020) or transformer (Kuznedelev et al., 2022; Kurtic et al., 2022) architectures. Moreover, second-order methods have shown some promise in pruning benchmarks specifically chosen to "fail current sparse neural networks" (Liu et al., 2023).

Despite the promise of OBS-derived approaches, they are faced with a severe tradeoff between scalability and accuracy that has proven hard to navigate. Specifically, both the importance scores

and the weight updates rely on estimating the action of the inverse Hessian $H^{-1}$ (or, in our case, the inverse Fisher matrix $F^{-1}$) on a high-dimensional parameter space ($\boldsymbol{v} \mapsto H^{-1}\boldsymbol{v}$), which inevitably calls for approximations. Indeed, all recent applications of the OBS framework to pruning have had to make significant simplifications, such as (i) ignoring correlations between most weights or groups of weights (Kurtic et al., 2022; Kuznedelev et al., 2022), or (ii) making low-rank approximations to the Hessian (Singh & Alistarh, 2020; Frantar et al., 2021) which are as good as the memory they consume. Note that these computational challenges also arise in second-order *optimization*.

In this work, we introduce the FishLeg surgeon (FLS) — a novel pruning algorithm that exploits the inverse curvature estimation machinery of the Fisher-Legendre (FishLeg) optimizer (Garcia et al., 2023). FishLeg attacks the scalability-accuracy dilemma by learning to directly amortize $F^{-1}\boldsymbol{v}$ products in an easy-to-evaluate $Q(\boldsymbol{\lambda})\boldsymbol{v}$ form. This is done by minimizing an auxiliary loss $\mathcal{A}(\boldsymbol{\lambda})$ derived from Legendre duality principles, w.r.t. a set of auxiliary parameters $\boldsymbol{\lambda}$ (details in Section 2). In contrast to low-rank approximations of the Fisher matrix that require hundreds of gradients to be stored, FishLeg allows the progressive distillation of a large number of gradients into the auxiliary parameter set $\boldsymbol{\lambda}$.[1] By means of low-parameter tensor factorization techniques, the size of $\boldsymbol{\lambda}$ can be kept within a small multiple of the size of the model itself, enabling pruning of large models with limited memory. Whilst such memory efficiency can also be attained through KFAC-based methods (Martens & Grosse, 2015; Wang et al., 2019), FishLeg's direct estimation of the *inverse* Fisher is less sensitive to gradient noise (Appendix G). Moreover, the form of KFAC's $F^{-1}$ follows rigidly from approximate mathematical derivations, whereas FishLeg's $Q(\boldsymbol{\lambda})$ can be any user-specified positive-definite quadratic form, yielding greater flexibility and accuracy. We use this flexibility to develop a novel variation on the well-known Kronecker-factored curvature approximation for dense layers, as well as new approximations for the convolutional layer.

We extend FishLeg's inverse Fisher estimation algorithm in a number of ways: (i) we modify the auxiliary loss $\mathcal{A}(\boldsymbol{\lambda})$ to facilitate assessment of its convergence and to promote learning of the full $F^{-1}$ as required for pruning (as opposed to learning the action of $F^{-1}$ on the subspace of momentary noisy gradients, as relevant to the optimization setting of Garcia et al., 2023); (ii) we propose a new preconditioner for this (often ill-conditioned) auxiliary loss, and show analytically that it accelerates convergence asymptotically; and (iii) we propose a new initialization scheme for $Q(\boldsymbol{\lambda})$ that leads to better estimation of $F^{-1}$ especially when it is ill-conditioned. We show that the FishLeg surgeon results in highly effective pruning on a number of benchmarks. We first demonstrate substantial one-shot and gradual pruning improvements over previous second-order methods on a deep autoencoder previously studied in the context of second-order optimization; this network is known to exhibit pathological curvature, and our results, therefore, suggest that FLS's superior inverse curvature estimator is key to improving pruning performance. Next, we apply FLS to the pruning of ResNet50. We show that in the one-shot and one-shot plus fine-tuning setup we consistently outperform other state-of-the-art second-order pruning methods, such as M-FAC and oBERT. Finally, although we do not explore this here, our method should be readily applicable to network quantization, following the approach traced out by Frantar et al. (2023).

## 2 BACKGROUND AND RELATED WORK

**Unstructured vs. structured pruning** Unstructured pruning reduces the number of parameters by scoring, and subsequently perhaps removing, each weight independently. In contrast, structured pruning scores and prunes entire components of the model, such as neurons, filters (Li et al., 2016), channels (He et al., 2017), layers (Fan et al., 2019; Sridhar & Sarah, 2020; Sajjad et al., 2023), or attention heads (Michel et al., 2019; Voita et al., 2019). Structured pruning therefore relies on an implicit structural understanding of the model. Recently, semi-structured pruning methods have gained popularity, where smaller subsets, e.g. blocks, of weights are removed together to allow the targeted hardware to take maximum advantage of sparsity (Lagunas et al., 2021; Gordon et al., 2020; Kurtic et al., 2022). Here, for simplicity, our experiments focus exclusively on unstructured pruning, but our method could easily be applied to the structured pruning case too.

---

[1]Although we do not explore this here, this direct and gradual learning of $F^{-1}$ in $Q(\boldsymbol{\lambda})$ is particularly relevant to the gradual pruning setting, where other methods typically have to recompute $F$ from scratch following pruning and re-invert it.

**One-shot pruning vs. gradual pruning**   One-shot pruning is the challenging task of pruning a model to some target sparsity with a single pruning iteration, and with no opportunity to recover any accuracy lost to pruning by e.g. re-training. In gradual pruning, the weights are instead removed progressively: each pruning step achieves some scheduled increase in sparsity and is followed by a period of fine tuning. This approach often allows higher sparsity to be obtained for a given model accuracy; for example, Gradual Magnitude Pruning (GMP; Gale et al., 2019; Han et al., 2016) often provides a strong baseline. M-FAC (Frantar et al., 2021) and oBERT (Kurtic et al., 2022) are relevant methods providing state-of-the-art results in second-order pruning in both pruning setups.

**Upstream vs. downstream pruning**   Downstream compression directly prunes while fine-tuning on a specific downstream task, Movement Pruning (Sanh et al., 2020) is an example. Alternatively, it is possible to compress the model upstream on the pre-training task as in Zafrir et al. (2021), significantly reducing the computational requirements because downstream fine-tuning on the pruned model requires training only a fraction of the initial set of weights. Some pruning methods (Kurtic et al., 2022; Frankle & Carbin, 2018) can be used in both upstream and downstream pruning.

**Fisher information matrix**   Consider a neural network with weights $\boldsymbol{w}$ that parameterize a predictive density $p(\boldsymbol{y}|\boldsymbol{x}, \boldsymbol{w})$ over labels $\boldsymbol{y}$ conditioned on the input $\boldsymbol{x}$. Let $\ell(\boldsymbol{w}, \{\boldsymbol{x}, \boldsymbol{y}\}) \triangleq -\log p(\boldsymbol{y}|\boldsymbol{x}, \boldsymbol{w})$ be the negative log-likelihood associated with data $\mathcal{D} = \{\boldsymbol{x}, \boldsymbol{y}\}$ (this will typically be the network's loss). The Fisher information matrix is defined as

$$F(\boldsymbol{w}) \triangleq \mathbb{E}_{\mathcal{D} \sim p(\mathcal{D}|\boldsymbol{w})}[\nabla_{\boldsymbol{w}}\ell(\boldsymbol{w}, \mathcal{D})\nabla_{\boldsymbol{w}}\ell(\boldsymbol{w}, \mathcal{D})^{\top}] \tag{1}$$

where $p(\mathcal{D}|\boldsymbol{w})$ is the model distribution (not the data distribution; typically, $p(\mathcal{D}|\boldsymbol{w}) \triangleq p(\boldsymbol{y}|\boldsymbol{x}, \boldsymbol{w})p^{\star}(\boldsymbol{x})$ where $p^{\star}(\boldsymbol{x})$ is the input data distribution)(Rao, 1992). We will denote by $\hat{F}$ any finite-sample approximation of $F$.

**Second-order pruning: OBS-based methods**   Most second-order pruning methods are based on the Optimal Brain Surgeon (OBS; Hassibi & Stork, 1992). OBS begins with a quadratic approximation of the loss function around the pre-trained parameter set $\boldsymbol{w}^{\star}$, typically assumed to be a minimum of the loss,

$$\delta\mathcal{L}(\delta\boldsymbol{w}) \quad \triangleq \quad \mathcal{L}(\boldsymbol{w}^{\star} + \delta\boldsymbol{w}) - \mathcal{L}(\boldsymbol{w}^{\star}) \quad \approx \quad \frac{1}{2}\delta\boldsymbol{w}^{\top}H(\boldsymbol{w}^{\star})\delta\boldsymbol{w}, \tag{2}$$

where $H(\boldsymbol{w}^{\star})$ is the Hessian of the loss at $\boldsymbol{w}^{\star}$. Here, we will approximate the Hessian by the Fisher $F(\boldsymbol{w}^{\star})$; most other works use the empirical Fisher matrix instead. This quadratic approximation leads to an analytical solution to the problem of optimally compensating for the deletion of a given weight $w_i$:

$$\delta\boldsymbol{w}^{\star} = -\frac{w_i^{\star}}{[F^{-1}(\boldsymbol{w}^{\star})]_{ii}}F^{-1}(\boldsymbol{w}^{\star})\boldsymbol{e}_i \tag{3}$$

where $\boldsymbol{e}_i$ is the $i^{\text{th}}$ canonical basis vector (Hassibi & Stork, 1992). The corresponding (minimal) increase in loss resulting from the deletion of weight $w_i$ is taken as its importance score:

$$\rho_i = \frac{w_i^2}{2[F^{-1}(\boldsymbol{w}^{\star})]_{ii}}. \tag{4}$$

These equations have also been extended to handle the semi-structured pruning setting whereby small blocks of weights are treated as single units (Kurtic et al., 2022).

Existing second-order pruning methods mostly differ in the way they estimate $F^{-1}\boldsymbol{v}$ products to compute Equations 3 and 4. All scalable methods make a block-diagonal approximation for $F$. WoodFisher (Singh & Alistarh, 2020) and oBERT (Kurtic et al., 2022) partition the parameter space into small blocks assumed to be independent, and use the Woodbury identity to recursively update an estimate of the inverse empirical Fisher $\hat{F}_{\mathcal{B}}^{-1}$ for each block $\mathcal{B}$. These approaches have substantial memory requirements ($\mathcal{O}(|\mathcal{B}|n)$, where $|\mathcal{B}|$ is the block size and $n$ is the total number of parameters in the model). M-FAC (Frantar et al., 2021) modifies this recursion to operate directly on $\hat{F}_{\mathcal{B}}^{-1}\boldsymbol{v}$ products, in a way that obviates the need for storing $\hat{F}_{\mathcal{B}}^{-1}$ (some parts of the computation can be cached and reused for any $\boldsymbol{v}$). This is typically much slower but requires less memory. In our work, FLS too approximates $F^{-1}$ in block-diagonal form, but with much larger blocks corresponding to entire layers, and with blocks structured to guarantee computational and memory efficiency.

**FishLeg**   FishLeg (Garcia et al., 2023) is a scalable second-order optimizer that approximates the natural gradient $F^{-1}\nabla_{\boldsymbol{w}}\ell(\boldsymbol{w},\mathcal{D})$ based on the following insights. Let $\boldsymbol{w}$ be a fixed set of model parameters. Consider the regularized cross entropy between $p(\mathcal{D}|\boldsymbol{w})$ and $p(\mathcal{D}|\boldsymbol{w}+\boldsymbol{\delta})$,

$$\mathcal{H}_{\gamma}(\boldsymbol{\delta}) = \mathbb{E}_{\mathcal{D}\sim p(\mathcal{D}|\boldsymbol{w})}\ell(\boldsymbol{w}+\boldsymbol{\delta},\mathcal{D}) + \frac{\gamma}{2}\|\boldsymbol{\delta}\|^2, \tag{5}$$

where $\gamma > 0$ is a small damping parameter. The Legendre-Fenchel conjugate of $\mathcal{H}_{\gamma}(\boldsymbol{\delta})$ is defined as

$$\mathcal{H}_{\gamma}^{\star}(\boldsymbol{u}) \triangleq \min_{\boldsymbol{\delta}} \mathcal{H}_{\gamma}(\boldsymbol{\delta}) - \boldsymbol{u}^{\top}\boldsymbol{\delta} \quad \text{with minimizer denoted by } \tilde{\boldsymbol{\delta}}_{\gamma}(\boldsymbol{u}). \tag{6}$$

Garcia et al. were able to prove that, if the negative log-likelihood $\ell(\boldsymbol{w},\mathcal{D}) = -\log p(\mathcal{D}|\boldsymbol{w})$ is twice differentiable, then the inverse damped Fisher information matrix exists and is equal to

$$F_{\gamma}^{-1} \triangleq [F + \gamma I]^{-1} = \nabla_{\boldsymbol{u}}\tilde{\boldsymbol{\delta}}_{\gamma}(\boldsymbol{0}). \tag{7}$$

FishLeg meta-learns a parametric approximation $\overline{\boldsymbol{\delta}}(\boldsymbol{u},\boldsymbol{\lambda})$ of $\tilde{\boldsymbol{\delta}}_{\gamma}(\boldsymbol{u})$, by minimizing the auxiliary loss $\mathcal{A}(\boldsymbol{\lambda},\boldsymbol{u}) \triangleq \mathcal{H}_{\gamma}(\overline{\boldsymbol{\delta}}(\boldsymbol{u},\boldsymbol{\lambda})) - \boldsymbol{u}^{\top}\overline{\boldsymbol{\delta}}(\boldsymbol{u},\boldsymbol{\lambda})$ w.r.t. meta-parameters $\boldsymbol{\lambda}$, as prescribed by Equation 6. Importantly, Equation 7 shows that one only needs to learn the *local* behaviour of the vector field $\tilde{\boldsymbol{\delta}}_{\gamma}(\boldsymbol{u})$ around small $\boldsymbol{u}$; thus, Garcia et al. directly parameterized its (symmetric, positive definite) Jacobian $Q(\boldsymbol{\lambda})$ at $\boldsymbol{u} = \boldsymbol{0}$, corresponding to the choice $\overline{\boldsymbol{\delta}}(\boldsymbol{u},\boldsymbol{\lambda}) \triangleq Q(\boldsymbol{\lambda})\boldsymbol{u}$. Furthermore, considering the limit of small $\boldsymbol{u}$ and averaging over a relevant distribution (more on this below and in Appendix E), the auxiliary loss becomes

$$\mathcal{A}(\boldsymbol{\lambda}) \triangleq \mathbb{E}_{\boldsymbol{u}}\left\{\frac{1}{\|\boldsymbol{u}\|^2}\left[\frac{1}{2}\boldsymbol{u}^{\top}Q(\boldsymbol{\lambda})F_{\gamma}Q(\boldsymbol{\lambda})\boldsymbol{u} - \boldsymbol{u}^{\top}Q(\boldsymbol{\lambda})\boldsymbol{u}\right]\right\} \tag{8}$$

which can be estimated and differentiated efficiently in a number of ways (details in Section 3).

Practical note: as $Q(\boldsymbol{\lambda})$ converges towards $F_{\gamma}^{-1}$, the auxiliary loss as defined by Equation 8 converges towards $\langle -\boldsymbol{u}^{\top}F_{\gamma}^{-1}\boldsymbol{u}/\|\boldsymbol{u}\|^2 \rangle$, which is problem-dependent; this makes it hard to assess the quality of our inverse Fisher estimation. We therefore assess convergence by computing a slightly modified auxiliary loss where we drop the $\frac{1}{2}$ factor; this should converge to zero.

Taking the gradient of Equation 8 w.r.t. $\boldsymbol{\lambda}$ makes is clear that $Q(\boldsymbol{\lambda})$ will learn to approximate the action of $F_{\gamma}^{-1}$ on the subspace spanned by the $\boldsymbol{u}$'s. Given their application to natural gradient optimization, Garcia et al. took those $\boldsymbol{u}$'s to be stochastic gradients of the model's primary loss function. For our pruning purposes, however, Equation 3 suggests that we must accurately estimate the action of $F$ on the entire parameter space; we will therefore work with a more isotropic distribution of $\boldsymbol{u}$ (Section 3).

Directly estimating the inverse Fisher matrix, and doing so in this way, brings a number of advantages. First, the FishLeg approach is flexible: one can specify any form of $Q(\boldsymbol{\lambda})$, and in particular combine structured approximations obtained through mathematical derivations (as in e.g. KFAC; Martens & Grosse, 2015; Grosse & Martens, 2016; George et al., 2018) with a variety of parametric adjustments for greater expressiveness. We give examples of such choices in Section 3.1. Second, the FishLeg approach is less biased than KFAC and related methods. These methods start by assuming that $F$ has a certain structure (e.g. block diagonal), obtain a good approximation of $F$ conforming to this structure, and then invert it. One expects both systematic errors as well as stochasticity in the estimate of $F$ to propagate to $F^{-1}$. In contrast, FishLeg 'fits' a parametric approximation to $F^{-1}$ directly, conveniently avoiding inversion. Relatedly, a key property of Equation 8 is that it is *not* biased by stochasticity in the estimate of $F_{\gamma}$ (Appendix G; Figure 4) – unlike other seemingly sensible auxiliary loss functions such as $\mathbb{E}_{\boldsymbol{u}}\|Q(\boldsymbol{\lambda})\hat{F}_{\gamma}\boldsymbol{u} - \boldsymbol{u}\|^2$ or $\mathbb{E}_{\boldsymbol{u}}\|\hat{F}_{\gamma}Q(\boldsymbol{\lambda})\boldsymbol{u} - \boldsymbol{u}\|^2$ whose quadratic terms in $\hat{F}_{\gamma}$ do survive averaging.

## 3   FISHLEG PRUNING

In this section, we describe the FishLeg surgeon, a novel application of FishLeg for pruning large neural networks within the OBS framework.

**One-shot pruning** of a pre-trained model with weights $\boldsymbol{w}^{\star}$ using FishLeg is described in Algorithm 1. We begin by learning to approximate the inverse Fisher $F_{\gamma}^{-1}(\boldsymbol{w}^{\star})$ by a positive definite matrix $Q(\boldsymbol{\lambda})$

parameterized in memory-efficient form (Section 3.1). We do so by minimizing FishLeg's auxiliary loss function (Equation 8) w.r.t. $\boldsymbol{\lambda}$. Whilst Garcia et al. (2023) estimated the auxiliary loss function and its gradient by sampling $\boldsymbol{u}$ as a gradient of the network's loss on some data minibatch, here we take $\boldsymbol{u}$ to be sampled from a standard Gaussian distribution. This promotes learning the full $F_\gamma^{-1}$, as opposed to learning its action on a restricted subspace dominated by the average gradient (when it is not exactly zero perhaps due to incomplete model training).

Following auxiliary loss minimization, we follow the standard OBS recipe. We score each weight using Equation 4 and select the bottom $f\%$ for deletion, where $f$ is the target sparsity. We then prune each of these weights by applying the OBS update of Equation 3. For this, we make the simplifying assumption that following deletion of weight $w_i$, the new (damped) inverse Fisher $F_\gamma^{-1}$ is identical to the old one, except for the removal of its $i^{\text{th}}$ row and column. Operationally, this allows us to prune all the selected weights at once (i.e. update a weight mask), and apply the update of Equation 3 restricted to the remaining weights. We speculate that better pruning could be obtained by proceeding more gradually, periodically updating $F_\gamma^{-1}$ (by resuming the minimization of the auxiliary loss) between pruning steps. We have not explored this here, mainly because the methods we compare FLS to do not update their curvature estimates in the one-shot setting either.

**Gradual Pruning** involves prune gradually in steps of increasing sparsity, with additional fine-tuning between each step. As opposed to pruning to the desired sparsity level in one shot, methods of gradually pruning a network are typically the best-performing pruning approaches. A so-called sparsity schedule specifies the sparsity level to prune to at each step (see e.g. grey curve in Figure 2, right). Critically, gradual second-order pruning requires re-estimation of the inverse FIM following each intermediate pruning step, to take into account the new masked parameters. Here, we reason that FishLeg's parametric estimation of the inverse FIM, $Q(\boldsymbol{\lambda})$, can be actively updated in a rolling fashion between consecutive pruning steps by simply performing a certain number of auxiliary loss minimization steps. We do this concurrently with the fine-tuning steps (for which we use the FishLeg optimizer, also based on the running estimate $Q(\boldsymbol{\lambda})$), as outlined in Algorithm 2. Hence, unlike previous approaches to gradual second-order pruning, we need not re-estimate and re-invert the Fisher matrix from scratch after each pruning step – we simply refine our current estimate.

## 3.1 MEMORY EFFICIENT PARAMETERIZATION OF THE INVERSE FISHER APPROXIMATION

For scalability, we approximate $F^{-1}$ in block-diagonal form, with each layer contributing one block. Note that these blocks are orders of magnitude larger than the ones used in previous second-order approaches that implemented direct inversion (e.g. Kurtic et al., 2022 used blocks of size 50). Our choice of structure for $Q(\boldsymbol{\lambda})$ is slightly more constrained by our pruning objective than it is for the FishLeg optimizer: we require efficient evaluation of not only $Q\boldsymbol{v}$ products but also $\text{diag}(Q)$ (required in Equation 4). For dense layers with $n_i$ inputs and $n_o$ outputs, and therefore with $(n_i+1)n_o$ parameters including biases, we parameterize the corresponding inverse Fisher block as

$$Q(\boldsymbol{\lambda}) \triangleq D(LL^\top \otimes RR^\top)D \tag{9}$$

where $D$ is a diagonal matrix with $(n_i + 1)n_o$ parameters, $L \in \mathbb{R}^{n_o \times n_o}$ and $R \in \mathbb{R}^{n_i \times n_i}$ are two parameter matrices, and $\otimes$ denotes the Kronecker product. This construction is such that, for $V \in \mathbb{R}^{n_o \times n_i}$,

$$Q(\boldsymbol{\lambda})\text{vec}(V) = D \odot \text{vec}(LL^\top(V \odot \bar{D})RR^\top) \tag{10}$$

with the (unusual) convention that $\text{vec}(\cdot)$ vectorizes row-wise (corresponding to a no-copy reshape in numerical code), and $\odot$ denotes elementwise (Hadamard) product. Here, $\bar{D} \in \mathbb{R}^{n_o \times (n_i+1)}$ is the un-vectorized version of the diagonal of $D$. Similarly, $\text{diag}(Q) = \text{diag}(D)^2 \odot (\text{diag}(LL^\top) \otimes \text{diag}(RR^\top))$ can be evaluated efficiently, with $\text{diag}(LL^\top) = (L \odot L)(1,\ldots,1)^\top$. Note that the inclusion of $D$ makes it more expressive than the standard KFAC approximation which is limited to the Kronecker product. For completeness in Appendix H, we compare the above parameterisation with a pure diagonal parameterisation and also a more restrictive block diagonal structure similar to other second-order pruning methods (i.e. oBERT & MFAC).

For convolutional layers (conv2D), we follow a similar tensor factorization strategy. Filter parameters are tensors of dimensions $n_o$(output channels) $\times$ $n_i$(input channels) $\times$ $K$(kernel size). Whilst we could parameterize the inverse Fisher block as a 3-way Kronecker product, Grosse & Martens (2016)'s KFAC derivation for convolutional layers suggests lumping together the input and kernel-

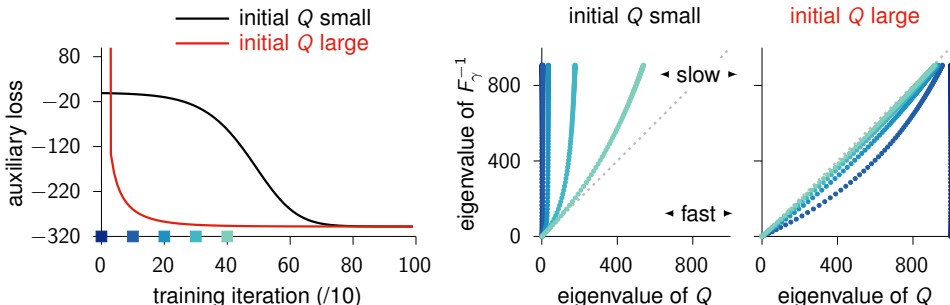

Figure 1: **The initialization of $Q(\boldsymbol{\lambda})$ matters much.** In this toy experiment, the true Fisher matrix ($n = 100$) was chosen so that its $i^{\text{th}}$ eigenvalue is $\xi_i \triangleq 1/i^2$, and the damping parameter $\gamma$ was set to $10^{-3}$. Thus, the eigenvalues of $F_\gamma^{-1}$ lie roughly in the $[1 - 1000]$ range. The auxiliary loss $\mathcal{A}(Q) = \frac{1}{2}\text{Tr}(QFQ) - \text{Tr}(Q)$ (left) was minimized by gradient descent w.r.t. the Cholesky factor of $Q(\boldsymbol{\lambda})$, initialized such that $Q(\boldsymbol{\lambda}) = I$ (black) or $Q(\boldsymbol{\lambda}) = \gamma^{-1}I = 1000 \times I$ (red). The learning rate was optimized separately for each case. This simulation shows that it is clearly better to initialize $Q$ to be large rather than small. Indeed, a simple derivation shows that each eigenvalue $\beta_i$ of $Q$ approaches its target $1/(\xi_i + \gamma)$ at a speed proportional to $(\xi_i + \gamma)$ (Equation 11). In other words, the eigenvalues of $Q$ that must end up large are also those that evolve the slowest. It, therefore makes sense to initialize them to be large so they have less to travel; the eigenvalues that must end up small will become small rapidly anyway. The right panels illustrate this behaviour by plotting the eigenvalues of $Q$ against their respective targets, at regular intervals during optimization (color-coded), for both initialization schemes. The auxiliary loss is minimized when $\beta_i = 1/(\xi_i + \gamma)$, i.e. when the dots lie along the identity line (dashed grey).

size dimensions. We therefore use the same structure as in Equation 9, but with $R$ of size $n_{\text{i}}K$ and $D$ of size $n_{\text{o}}n_{\text{i}}K$.

## 3.2 INITIALIZATION OF $Q$

Our experiments with FishLeg have revealed that the minimization of the auxiliary loss is very sensitive to initialization – to the point that getting it wrong can yield useless estimates of $F_\gamma^{-1}$. In the context of neural network optimization, Garcia et al. (2023) advocated an identity initialization $Q_0 = \alpha I$. To choose the value of $\alpha$, they observed that this identity initialization implied that the FishLeg update $\boldsymbol{w}_{t+1} \leftarrow \boldsymbol{w}_t - \eta Q(\boldsymbol{\lambda})\nabla_{\boldsymbol{w}}\mathcal{L}$ would initially correspond to SGD. Thus, given a learning rate $\eta_{\text{SGD}}$ known to work well for SGD, they set $\alpha \triangleq \eta_{\text{SGD}}/\eta$. However, in the context of pruning this rationale no longer applies; we therefore revisited the choice of $\alpha$.

We found that good pruning results could only be obtained for sufficiently large $\alpha$. To understand this, we studied the idealized dynamics of auxiliary loss gradient descent (Figure 1; see also Appendix F). Let $F = U\Xi U^\top$ be the eigendecomposition of the Fisher matrix, with $\Xi = \text{diag}(\xi_1, \ldots, \xi_n)$. Assuming $\boldsymbol{u} \sim \mathcal{N}(0, I_n)$, the auxiliary loss (Equation 8) reduces to $A(\boldsymbol{\lambda}) = \frac{1}{2}\text{Tr}(QF_\gamma Q) - \text{Tr}(Q)$. Expressing $Q$ in the eigenbasis of $F$ as $Q = U\beta U^\top$, the gradient flow for this deterministic loss function takes the form $\dot{\beta} = -(\Xi + \gamma I)\beta + I$ with $\beta(0) = \alpha I$. It is then easy to see that $\beta$ will remain diagonal throughout, and that the $i^{\text{th}}$ eigenvalue of $Q$ has the following dynamics:

$$\underbrace{(\xi_i + \gamma)^{-1}}_{\text{time constant}}\frac{d\beta_i}{dt} = -\beta_i + \underbrace{(\xi_i + \gamma)^{-1}}_{\text{optimal steady state}} \quad \text{with} \quad \beta_i(0) = \alpha. \tag{11}$$

Thus, the eigenvalues of $Q$ – all initially equal to $\alpha$ – converge at very different speeds depending on their optimal steady states: eigenvalues that must reach large (resp. small) values evolve slowly (resp. fast). We therefore conclude that a good initialization is to set $\alpha$ to be as large as the largest eigenvalues of $F_\gamma^{-1}$, namely $(\min\{\xi_i\} + \gamma)^{-1} \approx \gamma^{-1}$. This way, the eigenvalues of $Q$ that would normally slowly evolve towards $\gamma^{-1}$ are positioned there from the outset, and the eigenvalues that are set to decrease do so rapidly. Figure 1 illustrates this behaviour and shows that large initialization of $Q$ (with $\alpha \approx \gamma^{-1}$) results in faster minimization of the auxiliary loss.

### 3.3 PRECONDITIONING OF THE AUXILIARY LOSS

Learning the full $F^{-1}$ is a hard problem when $F$ is ill-conditioned, as the auxiliary loss inherits this ill-conditioning. Our theoretical analysis of this problem (Appendix F) has led to the discovery of a good preconditioner which only costs a single additional $Q(\boldsymbol{\lambda})v$ product per iteration (Algorithm 1). This preconditioner greatly accelerates asymptotic convergence of the auxiliary loss (Figure 5A), leading to better estimates of the inverse FIM.

## 4 EXPERIMENTS

The current state-of-the-art pruning approaches (Frantar et al., 2023; Kurtic et al., 2022; Liu & Wang, 2023) rely on sophisticated recipes, distillation strategies and quantization techniques to achieve a high level of model compression (Li et al., 2016). This work and the experiments below do not enter this level of specialization, focusing instead on (i) how using second-order pruning outperforms the first-order methods, (ii) how having a better approximation of the Fisher information matrix translates to better second-order importance scores and therefore to better pruning, (iii) and how by using FishLeg, we can learn more accurate Fisher matrix approximation for the most commonly used deep learning architectures, without resorting to complex approximations. FishLeg pruning can be combined with any of the methods cited above, as long as they are compatible with second-order pruning, to enhance their pruning capabilities further. Consequently, the experiments below start by revisiting the MLP-based autoencoder benchmark used in the FishLeg paper Garcia et al. (2023). We continue by exploring the pruning of one of the most famous CNN architectures, ResNet50 (He et al., 2015). For both experiments, we study the one-shot pruning and one-shot pruning with fine-tuning performance of our method and compare it with relevant baselines. For the autoencoder setup, we also investigate the gradual pruning performance.

The following subsections will discuss the experiments described above. All experiments are deployed on one RTX A6000 GPU with 48 Gigabytes GDDR6 RAM. In the interest of reproducibility, each experiment has been run five times with different initialisation seeds; in every figure, we show the mean of the five runs and the error bars corresponding to the standard deviation. If it is not possible to see the error bars, it is because they are too small and the results are therefore very consistent. For details of the experimental codebase see Appendix C.

### 4.1 MNIST AUTOENCODER

We first study second-order pruning with FishLeg in the MNIST autoencoder benchmark used in the FishLeg optimization paper to compare this algorithm with other second-order pruning methods. The architecture of the autoencoder is MLP-based and further details of its implementation can be found in (Goldfarb et al., 2020).

For all MNIST experiments, we prune a dense autoencoder model pre-trained via Adam on the same MNIST task that we use as target for pruning. In each case, the optimal hyperparameters were chosen via a grid search. The batch size is set at 100, and the network is optimized with respect to a negative log Bernoulli likelihood.

**One-shot pruning** We compare our algorithm against Global Magnitude Pruning and the SOTA second-order methods, oBERT and M-FAC. Both oBERT and M-FAC are set to collect 1024 gradients for their Fisher approximations, with their Fisher block sizes set to 50 and 2000 respectively (the default values in the SparseML codebase).

The results in Figure 2 show FLS consistently matching or outperforming the other baselines at all sparsity values. In higher sparsity regimes FLS is more robust than competing methods, with a 30% lower test loss at $90\%$ one-shot sparsity when compared with the next best method (oBERT).

**One-shot + Fine-tuning** In this section, we can observe how the FLS and the FishLeg optimizer work in tandem to prune and retrain using the same inverse Fisher approximation. In this experiment, we prune the model to 80% sparsity in one-shot and then fine-tune the sparse model for 20 epochs to recover as much performance as possible. Except where indicated, Adam is used as optimizer for fine-tuning with default PyTorch hyperparameters and weight decay set to $\lambda = 10^{-5}$.

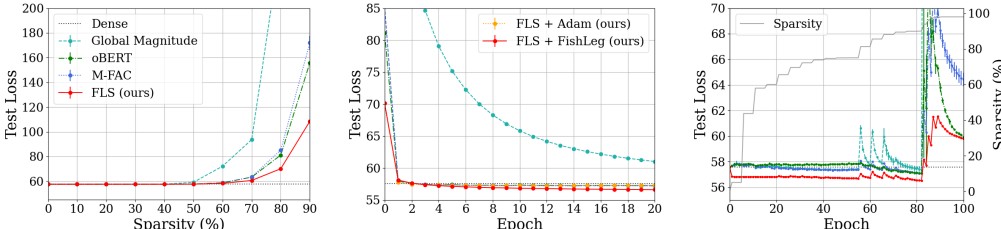

Figure 2: **MNIST autoencoder pruning test loss** (as negative log likelihood) for one-shot (left), one-shot + fine-tune at 80% sparsity (middle) and gradual pruning (right). The fine-tuning and gradual pruning are carried out using Adam optimizer, except for the FLS+FishLeg where the FishLeg optimizer is used to fine-tune. In all three experiments, FLS consistently outperforms the other baselines, especially at high sparsity values.

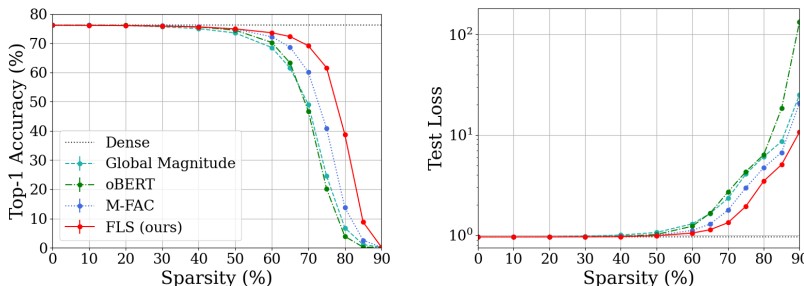

Figure 3: **ResNet50 performance on ImageNet after one-shot pruning** at different levels of sparsity. The top-1 accuracy metric (left) and the corresponding softmax test loss (right) as a function of sparsity are shown for each method.

From Figure 2 (middle), FLS fine-tuned with FishLeg optimizer is the only recipe that stands out and outperforms all other baselines. Note that the starting values of test loss shown at epoch 0 are the same values shown for one-shot pruning at 80% sparsity (see Figure 2, left) for each of the methods.

We can also observe that all second-order methods achieve a final lower loss than the original dense model after approximately 3 epochs. This phenomenon can be partially attributed to improved generalisation of the sparse model justified by Occam's hill (Blumer et al., 1987; Hoefler et al., 2021), such that the increase in performance can be explained by a reduction in learned noise.

**Gradual pruning** Figure 2 (right) shows the test loss as the gradual pruning progresses towards 98% sparsity (refer to the grey line and the right hand axis for sparsity schedule). FishLeg Surgeon consistently outperforms the other methods and shows to be more reliable at higher sparsities. All models seem to collapse above 90% sparsity, but the increase in test loss is significantly more contained for FLS compared to other methods.

We prune by using the estimate of the inverse Fisher from the FishLeg optimizer used for fine-tuning; this is a significant computational advantage compared to other second-order pruning methods. The Fisher approximation $Q(\boldsymbol{\lambda})$ is actively updated during the fine-tuning steps. As for other methods, Adam is used as optimizer for fine-tuning.

## 4.2   RESNET50

Scaling up to larger models and problem setups, we evaluate the FishLeg Surgeon performance on ResNet50 (He et al., 2016) pre-trained on ImageNet (Deng et al., 2009). The batch size is set at 128 and the model is optimized for classification with respect to the standard categorical likelihood.

Table 1: One-shot pruning + fine-tuning of ResNet50. Numbers denote test accuracies on ImageNet. ResNet50 is first pruned to $80\%$ sparsity in one-shot, and then fine-tuned for one epoch on the ImageNet dataset.

| Method | Top-1 Accuracy (%) | Top-5 Accuracy (%) |
|---|---|---|
| Dense | 76.14 | 92.87 |
| Global Magnitude | $70.44 \pm 0.12$ | $90.15 \pm 0.02$ |
| oBERT | $70.32 \pm 0.14$ | $90.14 \pm 0.02$ |
| M-FAC | $70.35 \pm 0.03$ | $90.14 \pm 0.09$ |
| FLS + Adam (ours) | $70.80 \pm 0.06$ | $90.42 \pm 0.06$ |
| **FLS + FishLeg (ours)** | **$71.91 \pm 0.08$** | **$90.78 \pm 0.04$** |

**One-shot pruning** For one-shot, we prune a ResNet50 model to various levels of sparsity up to 90%. These results are summarised in Figure 3 in terms of top-1 accuracy (left) and the resulting negative log-likelihood (right), both evaluated on ImageNet test data.

We show that the performance of FLS comfortably exceeds that of the baselines across all levels of sparsity, with 62% accuracy at 75% sparsity, compared to 41% for M-FAC and 24% for oBERT. In addition to this, we note that compared to the next best performing pruner (M-FAC, 42GB), FishLeg surgeon has $\times 2.4$ lower VRAM (17GB) consumption with our experimental setup.

**One-shot + Fine-tuning** Finally, we provide one-shot + fine-tuning results for ResNet50 with ImageNet, where the sparse model has been fine-tuned for one full epoch after pruning (Table 1). Whilst fine-tuning reduces the gap between FLS and other methods, FLS still yields the best performance, indicating that some of the one-shot improvements transfer to the fine-tuning regime too.

## 5 DISCUSSION, LIMITATIONS AND FUTURE WORK

We have shown that FLS is a computationally efficient algorithm achieving SOTA results for second-order pruning. FLS's advantages are especially significant at high sparsity, and our ablation experiments (Appendix H) suggest that they stem from a more accurate estimation of the inverse Fisher matrix. We speculate that the FishLeg machinery will benefit other applications that require accurate and tractable estimates of inverse curvature.

While the FishLeg pruning method is effective in many scenarios, it has several limitations. One of the key assumptions in our approach is that the inverse Fisher $F_\gamma^{-1}(\boldsymbol{w}^\star)$ can be well approximated by a specific form of positive definite matrix $Q(\boldsymbol{\lambda})$; however, the structure chosen for $Q$ is largely dictated by scalability requirements, and may not be appropriate under certain conditions. We have proposed memory-efficient factorizations of $Q$ which we have found effective for dense and convolutional layers, and we leave the development of other types of neural network layers to future research.

Another noteworthy assumption is that following deletion of weight $w_i$, the new (damped) inverse Fisher $F_\gamma^{-1}$ is the same as the old one, save for the removal of its $i^{\text{th}}$ row and column. This simplifying assumption is also used by previous approaches to pruning and leads to computational savings, but it potentially limits the accuracy of the pruning process (see the illustrative example given in Wang et al., 2019). To mitigate this, one could take advantage of the fact that FishLeg's auxiliary loss minimization enables gradual distillation of curvature information into $Q(\boldsymbol{\lambda})$. Maintaining an accurate running estimate of $F^{-1}$ as the model gets pruned is therefore less costly than with previous methods that typically require re-estimating and re-inverting $F$ from scratch following weight deletion. We have found this approach to be effective for gradual pruning of the MNIST autoencoder in Figure 2 (right), but leave further gradual pruning applications to larger networks for future work.

In conclusion, while the FishLeg pruning method represents a promising step forward in the efficient and effective pruning of large neural networks, the aforementioned limitations highlight directions for future improvements. Further research in these areas will likely extend and refine the capabilities of the proposed method.

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

## A  ONE-SHOT PRUNING

Algorithm 1 describes the details of the FishLeg surgeon algorithm for the one-shot pruning setup.

---

**Algorithm 1** FishLeg surgeon (one-shot setting)

---

1: **Goal**: One-shot pruning of network with $n$ parameters to $f\%$ sparsity.
2: Choose hyperparameters: damping factor $\gamma$, scale $\alpha$, Adam parameters.
3: Initialize $Q(\boldsymbol{\lambda}_0) = \alpha I_n$        ▷ *See section 3.2*
4: **while** not converged **do**
5:     Sample $\boldsymbol{u} \sim \mathcal{N}(0, I_n)$ from isotropic Gaussian distribution.
6:     $\boldsymbol{v} \leftarrow Q(\boldsymbol{\lambda})\boldsymbol{u}$        ▷ *Fast matrix-vector products*
7:     ▷ *Antithetic estimator of inverse FIM vector products*      ◁
8:     Sample one data minibatch $\mathcal{D}$ and estimate $\hat{F}\boldsymbol{v} = \frac{1}{2\epsilon}\big[\nabla_{\boldsymbol{w}}\mathcal{L}(\boldsymbol{w}+\epsilon\boldsymbol{v}, \mathcal{D}) - \nabla_{\boldsymbol{w}}\mathcal{L}(\boldsymbol{w}-\epsilon\boldsymbol{v}, \mathcal{D})\big]$
9:     `surrogate_loss` $\leftarrow \boldsymbol{v}^T$`stop_gradient`$(Q(\boldsymbol{\lambda}))\big[(\hat{F}+\gamma I)\boldsymbol{v} - \boldsymbol{u}\big]$ ▷ *With preconditioning*
10:     $\boldsymbol{\lambda} \leftarrow$ Adam_update$(\lambda, \nabla_{\boldsymbol{\lambda}}$`surrogate_loss`$)$.
11:     Assess convergence where `surrogate_loss` should approach zero.
12:
13: Compute the importance score $\rho_i$ of each weight $w_i$ using Equation 4, using the approximation $F^{-1}(\boldsymbol{w}^\star) \approx Q(\boldsymbol{\lambda})$.
14: Select and prune the $f\%$ least important weights (smallest $\rho_i$).
15: Apply the pruning update to the remaining weights using Equation 3.

---

## B  GRADUAL PRUNING

Algorithm 2 describes the details of the FishLeg surgeon algorithm for the gradual pruning setup.

---

**Algorithm 2** FishLeg surgeon (gradual-pruning setting)

---

1: **Goal**: gradual pruning to $f_{\text{end}}\%$ sparsity.
2: Choose hyperparameters: damping factor $\gamma$, learning rate $\eta$, Adam parameters, sparsity schedule $\{f_t\}$.
3: Pretrain $Q(\boldsymbol{\lambda}_0)$ using the same strategy as in Algorithm 1. ▷ *starts with a good estimate of the inverse FIM*
4: $t \leftarrow 0, \boldsymbol{w}_0 \leftarrow \boldsymbol{w}^*$
5: **while** not finished **do**
6:     ▷ *Pruning step*      ◁
7:     Select and prune the $(f_{t+1} - f_t)\%$ least important weights using the latest approximation $F_\gamma^{-1}(\boldsymbol{w}_t) \approx Q(\boldsymbol{\lambda}_t)$, where $\boldsymbol{w}_t$ is the masked parameters from the previous sparsity level.
8:     Use Equation 3 to obtain the new masked updated parameters $\boldsymbol{w}_{t+\frac{1}{2}}$.
9:     $\tilde{\boldsymbol{w}}_0 \leftarrow \boldsymbol{w}_{t+\frac{1}{2}}, \tilde{\boldsymbol{\lambda}}_0 \leftarrow \boldsymbol{\lambda}_t$
10:     **for** $s = 1 : S$ **do**
11:        $\mathcal{L}, \boldsymbol{g} \leftarrow$ value and gradient of loss evaluated at the masked $\tilde{\boldsymbol{w}}_s$ on a data minibatch
12:        ▷ *Fine-tuning*      ◁
13:        $\tilde{\boldsymbol{w}}_{s+1} \leftarrow \tilde{\boldsymbol{w}}_s - \eta[Q(\tilde{\boldsymbol{\lambda}}_s)\boldsymbol{g}]$      ▷ *masked update that preserves current sparsity*
14:        ▷ *Update the inverse FIM approximation, taking into account the new parameters*      ◁
15:        Perform one step of auxiliary loss minimization as in Algorithm 1, yielding a new $\tilde{\boldsymbol{\lambda}}_{s+1}$.
16:     ▷ *resume pruning with the fine-tuned parameters and updated inverse FIM estimate*      ◁
17:     $\boldsymbol{w}_{t+1} \leftarrow \tilde{\boldsymbol{w}}_S, \boldsymbol{\lambda}_{t+1} \leftarrow \tilde{\boldsymbol{\lambda}}_S, t \leftarrow t+1$

---

## C   FISHLEG SURGEON CODE

Our experimental code has been developed by introducing FishLeg surgeon in the Neural Magic SparseML library (Kurtz et al., 2020). The code is included with the supplementary material and will be made publicly available on acceptance of the paper.

As explained in Section 3.1, dedicated implementation of the $Q(\boldsymbol{\lambda})\boldsymbol{v}$ product for the convolutional layer has been developed as part of this work. This is an extension of the original FishLeg work as the original paper only considered the MLP case.

## D   EXPERIMENT HYPER-PARAMETER VALUES

Table 2 and Table 3 show the hyperparameter values used for each of the experimental setups, both for the FishLeg optimizer and the Adam optimizer. More details of the experiments can be found in Section 4.

Table 2: Optimal hyperparameter values for FishLeg, identified as the result of a grid search. These hyperparameters were chosen to minimise the training loss. Any parameters not shown are left as default values in the FishLeg optimizer library.

|  | 1-shot & 1-shot + FT | | Gradual |
| --- | --- | --- | --- |
|  | MNIST | IMAGENET | MNIST |
| Batch Size | 100 | 128 | 100 |
| $\eta$ | 1e-2 | 3e-4 | 2e-2 |
| $\alpha$ | 1e-5 | 1e-5 | 1e-5 |
| $\eta_{\text{aux}}$ | 1e-4 | 1e-5 | 1e-4 |
| $\beta$ | 0.3 | 0.9 | 0.7 |
| Damping $\gamma$ | 0.5 | 0.5 | 0.5 |
| Scale | 1.0 | 2.0 | 1.0 |
| Warmup | 1e4 | 1e3 | 1e4 |

Table 3: Optimal hyperparameter values for Adam, identified as the result of a grid search. These hyperparameters were chosen to minimise the training loss. Any parameters not shown are left as default values in the PyTorch Adam optimizer.

|  | 1-shot & 1-shot + FT | | Gradual |
| --- | --- | --- | --- |
|  | MNIST | IMAGENET | MNIST |
| Batch Size | 100 | 128 | 100 |
| $\eta$ | 1e-3 | 1e-4 | 1e-3 |
| $\alpha$ | 1e-5 | 1e-5 | 1e-5 |
| $[\beta_1, \beta_2]$ | $[0.9, 0.999]$ | $[0.999, 0.999]$ | $[0.9, 0.999]$ |

## E   AUXILIARY LOSS DERIVATION

Starting from the auxiliary loss definition given in Equation 8 and in Equation 15 of Garcia et al. (2023), we can expand the first term with a Taylor Expansion as:

$$\mathcal{H}_\gamma(\overline{\boldsymbol{\delta}}(\boldsymbol{u}, \boldsymbol{\lambda})) = \mathcal{H}_\gamma(\mathbf{0}) + \nabla_{\overline{\boldsymbol{\delta}}}\mathcal{H}_\gamma(\boldsymbol{\theta}, \overline{\boldsymbol{\delta}})|_{\overline{\boldsymbol{\delta}}=\mathbf{0}}\overline{\boldsymbol{\delta}} + \frac{1}{2}\overline{\boldsymbol{\delta}}^\top \nabla_{\overline{\boldsymbol{\delta}}}^2 \mathcal{H}_\gamma(\boldsymbol{\theta}, \overline{\boldsymbol{\delta}})|_{\overline{\boldsymbol{\delta}}=\mathbf{0}}\overline{\boldsymbol{\delta}}. \tag{12}$$

As stated in Appendix A.2 of Garcia et al. (2023), each term in this Taylor expansion can be expressed as:

$$\nabla_{\boldsymbol{\delta}}\mathcal{H}_\gamma(\boldsymbol{\theta}, \boldsymbol{\delta})|_{\boldsymbol{\delta}=\mathbf{0}} = \mathbb{E}_{\mathcal{D}\sim p(\mathcal{D}|\boldsymbol{\theta})}\nabla_{\boldsymbol{\theta}}\ell(\boldsymbol{\theta}, \mathcal{D}) + \mathbf{0} = \mathbf{0} \tag{13}$$

$$\nabla_{\boldsymbol{\delta}}^2\mathcal{H}_\gamma(\boldsymbol{\theta}, \boldsymbol{\delta})|_{\boldsymbol{\delta}=\mathbf{0}} = \mathbb{E}_{\mathcal{D}\sim p(\mathcal{D}|\boldsymbol{\theta})}\nabla_{\boldsymbol{\theta}}^2\ell(\boldsymbol{\theta}, \mathcal{D}) + \gamma I = \mathcal{I}(\boldsymbol{\theta}) + \gamma I = F_\gamma. \tag{14}$$

where the $0^{\text{th}}$ order term follows from the fact that we define the minimum at $\boldsymbol{\delta} = \mathbf{0}$, the $1^{\text{st}}$ order term is zero since we are at a minimum and the $2^{\text{nd}}$ order term characterizes the Fisher information matrix.

Using the above definitions, one can arrive at,

$$\mathcal{A}(\boldsymbol{\lambda}, \boldsymbol{u}) = \frac{1}{2}\overline{\boldsymbol{\delta}}(\boldsymbol{u}, \boldsymbol{\lambda})^{\top} F_{\gamma} \overline{\boldsymbol{\delta}}(\boldsymbol{u}, \boldsymbol{\lambda}) - \boldsymbol{u}^{\top}\overline{\boldsymbol{\delta}}(\boldsymbol{u}, \boldsymbol{\lambda}) \tag{15}$$

where the second term in Equation 8 is unchanged.

## F    ANALYSIS OF FISHLEG'S AUXILIARY LOSS & PRECONDITIONING

In this section, we analyze the minimization dynamics of a generalized version of FishLeg's auxiliary loss:

$$\mathcal{A}(Q) = \langle \frac{1}{2}\boldsymbol{u}^{\top}Q^{\top}PF_{\gamma}Q\boldsymbol{u} - \boldsymbol{u}^{\top}Q^{\top}P\boldsymbol{u}\rangle_{\boldsymbol{u}\sim\mathcal{N}(0,I)} \tag{16}$$

where $F$ is the model's (damped) Fisher information matrix, $P$ is a symmetric positive definite matrix, and $Q$ is our approximation of $F_{\gamma}^{-1}$. For simplicity, we will assume that the parameterization of $Q$ is non-limiting, i.e. we will consider the minimization of $\mathcal{A}$ directly as a function of $Q$.

This loss can be evaluated analytically:

$$\mathcal{A}(Q) = \left\langle \text{Tr}\left(\frac{1}{2}Q^{\top}PF_{\gamma}Q\boldsymbol{u}\boldsymbol{u}^{\top} - Q^{\top}P\boldsymbol{u}\boldsymbol{u}^{\top}\right)\right\rangle_{\boldsymbol{u}\sim\mathcal{N}(0,I)} \tag{17}$$

$$= \text{Tr}\left[\left(\frac{1}{2}Q^{\top}PF_{\gamma}Q - Q^{\top}P\right)\langle\boldsymbol{u}\boldsymbol{u}^{\top}\rangle_{\boldsymbol{u}\sim\mathcal{N}(0,I)}\right] \tag{18}$$

$$= \text{Tr}\left(\frac{1}{2}Q^{\top}PFQ - Q^{\top}P\right) \tag{19}$$

The optimal $Q^{\star}$ must satisfy

$$0 = \left.\frac{\partial\mathcal{A}}{\partial Q}\right|_{Q=Q^{\star}} = P(FQ^{\star} - I) \tag{20}$$

Therefore, if $P$ and $F_{\gamma}$ are both invertible, then $Q^{\star} = F_{\gamma}^{-1}$ as desired. To understand how quickly $Q$ will converge to this solution, it is useful to analyze the gradient flow

$$\frac{dQ}{dt} = -P(F_{\gamma}Q(t) - I) \tag{21}$$

with initial condition $Q(0) = \alpha I$. Let $F = U\Lambda U^{\top}$ be the eigendecomposition of the Fisher matrix, with $\Lambda = \text{diagm}(\lambda_1, \ldots, \lambda_n)$ and $U^{\top}U = UU^{\top} = I$. We will assume that $P$ has the same eigenvectors as $F$, i.e. $P = U\text{diagm}(p_1, \ldots, p_n)U^{\top}$. Rewriting the above gradient flow in the eigenbasis of $F$, we obtain

$$\frac{d}{dt}(U^{\top}Q(t)U) = -U^{\top}P(F_{\gamma}Q - I)U \tag{22}$$

$$= -U^{\top}U\text{diagm}(p_1, \ldots, p_n)U^{\top}(U(\Lambda + \gamma I)U^{\top}Q - I)U \tag{23}$$

$$= -\text{diagm}(p_1, \ldots, p_n)((\Lambda + \gamma I)U^{\top}QU - I) \tag{24}$$

We see that if $U^{\top}QU$ is diagonal at time $t$, it will remain diagonal. Given that $U^{\top}QU = U^{\top}(\alpha I)U = \alpha I$ is diagonal, we conclude that at any time $t$, $U^{\top}Q(t)U = \text{diagm}(\beta_1(t), \ldots, \beta_n(t))$. Thus, Equation 24 boils down to a set of $n$ decoupled, scalar flows,

$$\frac{d\beta_i}{dt} = -p_i\left[(\lambda_i + \gamma)\beta_i - 1\right] \quad \text{with } \beta_i(0) = \alpha \tag{25}$$

These equations are more easily interpreted when rewritten as

$$\frac{\beta_i^{\star}}{p_i}\frac{d\beta_i}{dt} = -\beta_i + \beta_i^{\star} \tag{26}$$

where $\beta_i^{\star} = (\lambda_i + \gamma)^{-1}$ is the corresponding eigenvalue of the solution $Q^{\star}$ (the "target eigenvalues"). The solution to these dynamics is

$$\beta_i(t) = \beta_i^{\star} + (\alpha - \beta_i^{\star})\exp\left(\frac{-t}{\tau_i}\right) \quad \text{with } \tau_i \triangleq \frac{\beta_i^{\star}}{p_i}. \tag{27}$$

For $P = I$, i.e. $p_i = 1$, we recover the result of the main text (c.f. Figure 1): $\beta_i$ converges exponentially to its target $\beta_i^\star$, but on a timescale $\tau_i$ proportional to $\beta_i^\star$ itself. This is a problem when $F_\gamma$ is poorly conditioned, such that there is a broad range of $\beta_i^\star$: in this case, some $\beta_i$'s will converge rapidly, and some others will converge very slowly.

Equation 27 suggests a solution based on a judicious choice of the preconditioner $P$. If somehow we could precondition the loss with $P = F_\gamma^{-1}$, then $p_i = \beta_i^\star$ and therefore $\tau_i = 1$ for all $i$ – this case we have rapid uniform convergence of the inverse Fisher in all directions. While we do not know $F_\gamma^{-1}$ (indeed this is what we are trying to learn ...), we do know that $Q(t)$ is supposed to converge (albeit slowly) towards $F_\gamma^{-1}$. Thus, we propose a simple time-dependent preconditioner $P(t) = Q(t)$. Empirically, we do find that this choice leads to better asymptotic convergence of the auxiliary loss, as illustrated in Figure 5A. Note that this only costs a single additional $Qv$ product in every iteration.

## G  FISHLEG INVERSE CURVATURE ESTIMATION: FLEXIBLE AND ACCURATE

In this section, we report on a series of simple experiments that show that FishLeg's inverse curvature estimation is typically more accurate and flexible than more conventional approaches.

First, Figure 4A shows that – when the parameterization of FishLeg's $Q$ is sufficiently expressive to include $F_\gamma^{-1}$, $Q$ converges to $F_\gamma^{-1}$ as desired, despite only having access to stochastic estimates of $F$. This is because, using standard unbiased estimates of the Fisher matrix (or, practically, Fisher-vector products) on mini-batches in Equation 8, FishLeg's auxiliary loss and its gradient are also unbiased. With sufficiently small learning rate, we therefore expect $Q$ to converge to the inverse damped Fisher solution. In contrast, a more naive scheme that computes an average of inverses of noisy Fisher estimates ('est. – inv. – avg.' in Figure 4A) yields a bias that persists asymptotically.

Second, when $F_\gamma^{-1}$ lies outside the domain of the structured approximation (e.g. when it is not exactly a single Kronecker product, or a block-diagonal matrix), there is an advantage to directly approximating $F_\gamma^{-1}$ in the desired structured form $Q$ (FishLeg's strategy), rather than approximating $F$ in such a form and then inverting the result. For one, (Garcia et al., 2023) had already argued that the former is more flexible than the latter, because one can use structured forms that need not be easily inverted (indeed FishLeg does not invert anything). Here, we show that even when the structured form is easily inverted, FishLeg still has a marked advantage (Figure 4B-D). In particular, the auxiliary loss allows the specification of a distribution of vectors $u$ (specifically, their covariance) to promote learning the action of $F_\gamma^{-1}$ on select directions in parameter space. This is not possible in a more conventional approach whereby the Fisher matrix is first approximated in structured form, then averaged, and finally inverted.

## H  ADDITIONAL ABLATION EXPERIMENTS

For the experiments discussed in this section, a simple linear layer with $n$ inputs and a single output is used to perform controlled ablations and compare various approximations of the inverse Fisher and their impact on one-shot pruning. In Figure 5A-C we choose $n = 100$ and in Figure 5D we set $n = 500$. The layer weights are drawn from $\mathcal{N}(0, 1/n)$, and inputs are drawn from $\mathcal{N}(0, \Sigma_x)$, where $\Sigma_x$ is a random covariance matrix with eigenvalues $\{\lambda_i \propto e^{-i/10}\}$. Results are reported as mean $\pm$ s.e.m. over random seeds. Across all experiments, a batch size of 100 is chosen along with a damping parameter $\gamma = 0.01$. Note that in this toy example, the Fisher matrix is $F = \Sigma_x$, and does not depend on the weights. Figure 5A shows the effect of preconditioning the FishLeg auxiliary loss using the momentary approximation $Q(\lambda)$ of the inverse Fisher matrix. We observe that this preconditioning does indeed lead to faster asymptotic convergence. This is shown here for the 'full' approximation $Q = LL^\top$, which – in this case – is as expressive as the Kronecker parameterization of dense layers we have used in the experiments from the main text.

Figure 5B displays the quality of approximation of the inverse damped Fisher matrix, as measured by FishLeg's auxiliary loss after convergence[2], for various parameterizations of $Q(\lambda)$. We compare the 'full' parameterization $Q = LL^\top$ (orange), a positive diagonal parameterization (purple), and a set of positive-definite block-diagonal approximations with various block sizes (blues). These results show

---

[2]Where the Adam learning rate separately tuned for each approximation.

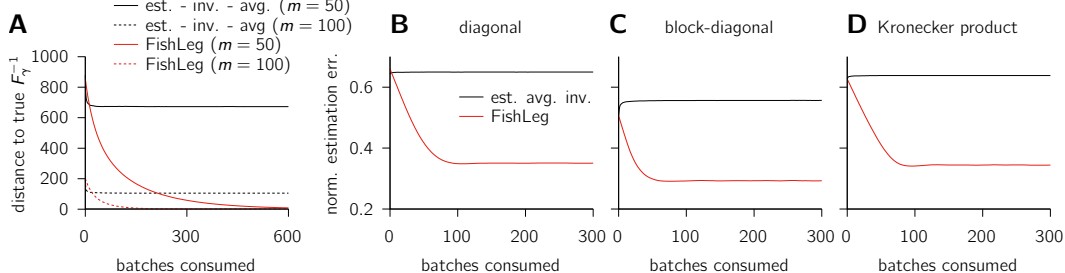

Figure 4: **Assessing FishLeg's inverse curvature estimation in a controlled setting**. In this figure, the true Fisher matrix $F \in \mathbb{R}^{100 \times 100}$ is constructed to have a random orthonormal eigenbasis and eigenvalues $\lambda_i \propto e^{-i/30}$. All results are averaged over 20 independent realizations of the corresponding experiment with different random seeds. **(A)**: standard affine-invariant Riemannian distance between FishLeg's $Q$ and $F_\gamma^{-1}$ ($\gamma = 0.01$), as a function of the number of data mini-batches of size $m$ consumed so far. Each Adam step of auxiliary loss minimization consumes one minibatch. In this case, we use a full parameterization $Q = LL^\top$ that contains the solution $F_\gamma^{-1}$; in that case, FishLeg's inverse curvature estimation is consistent and the error goes to zero. As a baseline, we show the behaviour of a simple but biased estimator that estimates $F_\gamma$ on each new minibatch, inverts that noisy estimate, and averages the result over minibatches; inverting noisy estimates yields a bias that persists asymptotically. **(B-D)**: In these panels, the inverse Fisher is estimated in structured form (B: diagonal; C: block-diagonal, 5 blocks; D: Kronecker product, $(5 \times 5) \otimes (20 \times 20)$. This is done either by FishLeg assuming a correspondingly structured form for $Q$ (red), or by (i) approximating $F_\gamma$ in structured form for each minibatch (for the Kronecker approximation, we use a permuted SVD to find the nearest Kronecker product in the least-squares sense; Van Loan & Pitsianis, 1993), (ii) averaging over minibatches (for the Kronecker approximation the two factors are averaged separately, as in KFAC), and (iii) inverting the result (black; note that in this case, the inverse inherits the structure). We report the squared error between $Qu$ and $F_\gamma^{-1}u$, averaged over $u \sim \mathcal{N}(0, \Sigma_u)$, and normalized by the average norm of $F_\gamma^{-1}u$. Here, to reflect the need of accurately estimating the action of $F_\gamma^{-1}$ on the least salient parameter dimensions, we have chosen $\Sigma_u = F^{-1}$.

very clearly that a full approximation can achieve a much lower auxiliary loss when compared to less powerful approximations in this case.

Following from this, Figure 5C is reporting the one-shot pruning performance (test MSE) for the various FishLeg parameterizations shown in Figure 5B, as well as for magnitude pruning (black), MFAC ($m = 10$; green) and 'exact FLS' with $F = \Sigma_x$ appropriately masked and inverted before each pruning step (red). One can observe that the full approximation achieves a far closer performance to the 'exact' result across all other baselines in this study. Note that in this case, the 'exact FLS' characterises the limit of performance for second-order pruning methods. In this setting, we therefore find a strong correlation between the quality of the iFIM approximation (as measured by Garcia et al. (2023)'s auxiliary loss after convergence) and one-shot pruning performance (comparing Figure 5B and Figure 5C). In particular, block-diagonal approximations (as used by OBS/oBERT) perform worse than the Kronecker-factored approximation (in this case also exact) and, indeed not much better than magnitude pruning or a simple diagonal approximation of the iFIM. Likewise, FLS with a Kronecker-factored Q performs better than MFAC (with rank parameter $m$ generously set to 10, i.e. 10% of the parameter count, which would normally be intractable memory-wise).

Finally, Figure 5D provides a comparison between FLS with block-diagonal parameterization and oBERT for various block sizes (5, 10, 20, 50). In particular, this ablation study shows benefits of directly estimating the inverse FIM than estimating the FIM and inverting it. oBERT utilizes the WSM formula for effective estimation without explicit inversion, resulting in iterative update of the inverse of moving average for the empricial Fisher matrix. In the top panels, we present one-shot pruning performance (test MSE) as a function of sparsity for the two methods. In the middle panels, the standard affine-invariant Riemannian distance between the masked approximate block-diagonal inverse and the true masked Fisher inverse are shown, for each method. In the bottom panels, the wall-clock time as a function of sparsity is shown. For these experiments, oBERT uses 512 gradients

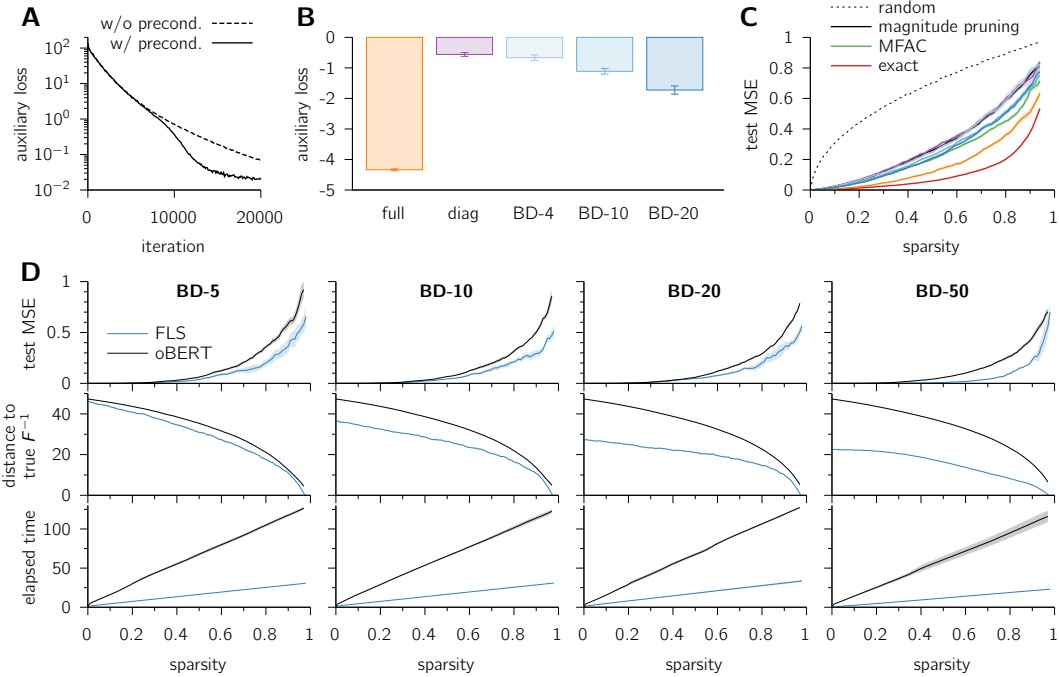

Figure 5: **Ablation experiments on synthetic data** in a toy setup to show: (A) the utility of preconditioning the auxiliary loss, (B) the predicted quality of the approximated Fisher in different scenario's, (C) the one-shot pruning performance of various Fisher approximations (including other baselines) and (D) the effect of implementing a block diagonal FishLeg approximation and it's comparison to oBERT at various block sizes.

at each pruning step, whereas FLS performs 20 steps of auxiliary loss minimization between pruning updates. These results show a systematic improvement in the inverse FIM estimates when using FLS, which implies that directly approximating the inverse Fisher in block-diagonal form (FLS) is better than approximating the Fisher in block-diagonal form before inverting each block (oBERT).

