# OpenReview forum: "Pruning neural networks using FishLeg estimation"
_ICLR.cc/2024/Conference — Submitted to ICLR 2024_

### Official Review · Reviewer_fDy7 · 2023-10-24

**Soundness:** 2 fair
**Presentation:** 3 good
**Contribution:** 1 poor
**Rating:** 3
**Confidence:** 4

**Summary:**

The authors propose a novel method of deep neural network unstructured pruning (sparsity). They claim that the best-performing pruning techniques use second-order methods for importance estimation. However, due to the size of modern neural networks, these methods are computationally too expensive. To address this limitation, the authors introduce FishLeg surgeon (FLS). The core idea is to leverage an accumulation of the gradients instead of storing them individually. This is achieved through tensor decomposition for an effective approximation. The authors mainly evaluate the proposed method on ResNet 50 trained on ImageNet.

**Strengths:**

Deep neural network compression is of paramount importance for future deployment. The authors proposed a novel method which brings marginal improvements over previous state-of-the-art methods.

**Weaknesses:**

I see three major concerns with this work
1. the empirical validation is not sufficient for a conference like ICLR. Research on pruning should at least involve a transformer architecture in its benchmarks. This has been the case for a few years in quantization.
2. The current results on ResNet 50 suggest that the benefits of the proposed method in terms of accuracy v.s. compression trade-offs are marginal and do not include many other works such as [1,2] which all achieve more impressive results (without using second order importance estimation)
3. The authors list many advantages of FishLeg which translate in marginal improvements on ImageNet

[1] Wimmer, Paul, Jens Mehnert, and Alexandru Condurache. "Interspace pruning: Using adaptive filter representations to improve training of sparse cnns." Proceedings of the IEEE/CVF conference on computer vision and pattern recognition. 2022.

[2] Yvinec, Edouard, et al. "Singe: Sparsity via integrated gradients estimation of neuron relevance." Advances in Neural Information Processing Systems 35 (2022): 35392-35403.

**Questions:**

I have listed a few concerns above. I will wait for the authors' response regarding 2 and 3. With respect to 1, i would like to open a discussion with other reviewers.

---

> ### Author Response · Authors · 2023-11-22
>
> We appreciate your thoughtful review. We have addressed your concerns in our main rebuttal above, where we address common questions among different reviews.

---

### Official Review · Reviewer_vdNS · 2023-10-28

**Soundness:** 2 fair
**Presentation:** 2 fair
**Contribution:** 1 poor
**Rating:** 3
**Confidence:** 4

**Summary:**

This paper proposes FishLeg Surgeon (FLS), a pruning technique that uses the FishLeg under the hood (an already existing approach to estimate Fisher Information Matrix (FIM) in neural networks, initially used for optimization) in the Optimal Brain Surgeon (OBS) framework to perform unstructured pruning, more precisely one-shot and gradual pruning. The experiments are performed on an autoencoder for MNIST and ResNet-50 on ImageNet.

**Strengths:**

1. the paper is easy to follow
2. the results on the AutoEncoder and ResNet-50 show improvements over other approaches, such as M-FAC and oBERT at 70%, 80% and 90% sparsity

**Weaknesses:**

I believe that paper contribution is not good enough for ICLR standards. Since the authors adapted the FishLeg implementation for pruning, I would have expected a broader evaluation process. The AutoEncoder benchmark on MNIST (which is also used in the FishLeg paper) and ImageNet on ResNet-50 are not that relevant for pruning.

The M-FAC and oBERT baselines are not state of the art for ImageNet/ResNet-50 benchmark. For example, in the Figure 8 from the CAP approach [1] also show around 70% accuracy for M-FAC for the same benchmark (ImageNet/ResNet-50 @ 75% sparsity), but the CAP approach is much better than M-FAC, reaching about 75% accuracy for 75% sparsity.

The paper does not have any experiments on LLMs pruning. Since this paper addresses the one-shot pruning too, some good results can be obtained using SparseGPT [2] technique, which showed good results on one-shot pruning on large models.


References:

[1] **CAP: Correlation-Aware Pruning for Highly-Accurate Sparse Vision Models**, available at **https://arxiv.org/pdf/2210.09223.pdf**

[2] **SparseGPT: Massive Language Models Can be Accurately Pruned in One-Shot**, available at **https://openreview.net/pdf?id=gsP05g8IeK**

**Questions:**

Given the presented weaknesses, I would like to add the following questions and I would appreciate if you could answer them one by one.

1. did you run M-FAC and oBERT from scratch during your evaluation process?
2. how does FLS behave for other tasks, such as:
- LLMs pruning, such as BERT on GLUE/SQuAD
    - for example, against gradual pruning on oBERT
    - one-shot pruning on large models against SparseGPT
- ViT or DeiT on ImageNet against CAP

---

> ### Public Comment · ~Jon_Crowcroft1 · 2023-11-17
> **LLM pruning & comment on CAP**
>
> There are no established metrics for accuracy for LLMs so you can't really measure the decrease in "accuracy" for pruning levels - the SparseGPT paper uses perplixity as a proxy, which is a short range predictive text NLP metric, and largely inappropriate for capturing the effects of attention, or indeed the high variance in errors sometimes called Hallucinations are not captured by such a limited measure, and not great for saying what the pruning tradeoff with accuracy...
>
> The CAP paper is nice and we should definitely add a comparison
>
> The SInGE pape uses integradted gradients as a good metric for accuracy (basically any of the family of XAI techniques are candidates) but the cost of this scales badly, so again looking at larger systems like GPT would not be affordable, sadly - even a relatively lower cost technique like Shapley Values would make the evaluation beyond most peoples' computational resources, and not in the least bit sustainable.
>
> The Interspace pruning approach is very nice, although complimentary to FishLeg (and CAP)
>
> Of course there are quite a few other second order pruning approaches out there - one could spend forever comparing each and every one...

---

> ### Author Response · Authors · 2023-11-22
>
> We appreciate your thoughtful review. In our main rebuttal, we addressed the common concerns of all reviews, including yours. As discussed above, this work focuses on improving second-order importance scores and, therefore, pruning using FishLeg surgeon without entering into any specialization of pruning tactics (recipes and distillation strategies) or network architectures. Therefore, our experiment uses the same benchmarks as previous second-order methods [3]. Applying FLS to LLMs pruning is undoubtedly interesting but outside this project's scope.
> We want to highlight that FLS can be combined with any state-of-the-art pruning methods as long as they are compatible with second-order pruning, further enhancing their capabilities.

---

> ### Comment · Reviewer_vdNS · 2023-11-22
>
> Thank you for the reply. I do not agree with what you said above. Since you are proposing a new pruning technique, I believe you must compare with the existing state of the art results on the latest benchmarks. The fact that you refuse to perform these experiments makes me consider that my review is not addressed at all and as a consequence I will keep my score.

---

### Official Review · Reviewer_s1pu · 2023-10-31

**Soundness:** 2 fair
**Presentation:** 3 good
**Contribution:** 2 fair
**Rating:** 5
**Confidence:** 2

**Summary:**

The authors proposed a Optimal Brain Surgery pruning technique where importance is based on the Fisher information matrix.  Following the FishLeg optimization due to Garcia et al., they proposed a specific parameterization of the inverse Fisher information matrix, as well as optimization procedures.  They further demonstrated with experiments on unstructured network pruning.

**Strengths:**

A potentially better yet practically tractable importance measure for OBS is of value to network compression practice in general.  The hypothesis that an approximated Fisher information inverse by FishLeg meta-optimization can play such a role is a novel idea.

**Weaknesses:**

- The parameterization of the inverse of Fisher information matrix (Eq. 9) is not unique even under the practicality constraint.  There might exist a practical tradeoff between the capacity and form of the parameterization and the quality of the resulting importance metric for pruning.
- As the authors demonstrated, the procedure of meta-optimization of $\lambda$ has hyperparameters that are tricky to tune.  This leads to practical complexity.
- Lack of demonstration with large models in comparison against competing techniques.
- Even with the small-model examples presented, the superiority of the proposed method has not been convincingly demonstrated.  For example, if the proposed importance metric (Eq. 4) is indeed superior than that from a competing method, e.g. OBC, then it is necessary to show the disagreement between them with a concrete example, e.g. a specific layer in Resnet50, where the optimal solutions in one is suboptimal in the other, but the current solution leads to lower loss change.

**Questions:**

See above.

---

> ### Author Response · Authors · 2023-11-22
>
> We appreciate your thoughtful review. In the main rebuttal above, we have addressed the concerns common to all reviews, including your questions about the lack of demonstration with large models and the superiority of FLS compared to other second-order methods.
> With respect to your concerns about the non-uniqueness of the $Q(\lambda)$ parameterization of the inverse Fisher, we claim that this flexibility is in fact an advantage of our method, as it allows the user to trade off accuracy for scalability. Nevertheless, in this work, we propose concrete approximations for the dense and convolutional layers that allow us to achieve better pruning performance than other second-order computational methods while having lower computational costs, as described in the general answer above.
>
> This reviewer also has expressed concern about the possible practical complexity of finetuning the hyperparameters of meta-optimization of $\lambda$. We are the first to acknowledge that hyperparameter tuning was an issue that was left unaddressed by the original FishLeg paper, and we did in fact spend considerable time thinking of ways to mitigate this:
>
> - Section 3.2 shows that a good diagonal initialization of $Q$ should have a magnitude equal to the largest eigenvalues of $F_\gamma^{-1}$, which is roughly $1/\gamma$ (Figure 1).
>
> - Section 3.3 acknowledges that learning the full $F^{-1}$ is a complex problem when $F$ is ill-conditioned, as the auxiliary loss inherits this ill-conditioning. Nevertheless, our theoretical analysis of this problem (Appendix F) has led to the discovery of a good preconditioner which only costs a single additional $Q(\lambda)v$ product per iteration (Algorithm 1).
> This preconditioner greatly accelerates the asymptotic convergence of the auxiliary loss (Figure 5A), leading to better estimates of the inverse FIM.

---

### Official Review · Reviewer_pttL · 2023-11-02

**Soundness:** 3 good
**Presentation:** 3 good
**Contribution:** 3 good
**Rating:** 6
**Confidence:** 2

**Summary:**

This work proposes a novel pruning mechanism that uses the FishLeg optimizer, which is based on the the inverse fisher information matrix. This work proposes a number of improvements to the fishleg optimizer to make it more amenable for unstructured pruning, such as  modeling the full FIM, as opposed to its action of a parameter subspace, as well as a preconditioner for the auxiliary loss. When applied to pruning, the authors show that there are improvements on various benchmarks, outperforming other second-order methods, and shows potential for network quantization applications.

**Strengths:**

* Interesting application of fishleg optimizer
* Fishleg extended to model the full inverse FIM with preconditioning
* Efficient and flexible parameterization of inverse FIM
* Good experimental results on benchmarks (figures 2,3)  compared to approaches like oBERT and M-FAC

**Weaknesses:**

* Efficiency is mentioned as an important component of the method, but no timing analysis was performed, There is some mention of memory consumption, but this is not made concrete.
* The introduction claims that the largest models are inaccessible to those without compute resources. How does this method help this situation when only ResNet-50 (that anyone can run) is examined.
* Results only show small dense autoencoder and resnet-50, would be nice to see more architectures and tasks.
* Results are quite marginal for imagenet (table 1), but I acknowledge that competing approaches saw smaller gains over each other.

**Questions:**

1. What is the computation burden of all methods? ResNet-50 may take up < 3Gb of VRAM, so 17GB is quite a lot more.

---

> ### Author Response · Authors · 2023-11-22
>
> We appreciate your thoughtful review. We have addressed your concerns in our main rebuttal above, addressing common questions among different reviews.

---

### Author Response · Authors · 2023-11-22
**General response to all reviewers**

# General response to all reviewers
We thank all reviewers for their time and effort in reviewing our paper.
We have identified in the four reviews two common themes, and we are addressing them here jointly. More specific comments/questions are addressed separately for each reviewer.

## First Question: About broader evaluation process and comparison with State-of-the-art algorithm.
This work does not aim to compete with successful state-of-the-art pruning approaches that are usually architecture-dependent and rely on sophisticated recipes and distillation strategies, such as SparseGPT[1] or Interspace pruning [2]. Our method does not enter this level of specialization; instead, it focuses on improving second-order pruning using FishLeg. Indeed, our algorithm could be combined with any of the state-of-the-art methods cited above to enhance their pruning capabilities further.

Our experiments use the same benchmarks as the previous second-order pruning papers, such as Woodfisher et al. [3]. In this work, first, we develop the FishLeg surgeon algorithm (FLS): we show how we can use FishLeg to better approximate the Fisher matrix, which translates to better second-order importance scores and, consequently, better pruning. Therefore, in this paper, we use simple autoencoders to illustrate the theory behind FLS and to compare this algorithm with other second-order pruning methods. Then, we apply FLS to ResNet-50, the standard benchmark used in second-order pruning papers, showing improved performance (as shown in Figure 2 and Figure 3 of the paper) with lower computational cost (see next question).

[1] Elias Frantar, Dan Alistarh. "SparseGPT: Massive Language Models Can be Accurately Pruned in One-Shot".arXiv:2301.00774. 2023
[2] Wimmer, Paul, Jens Mehnert, and Alexandru Condurache. "Interspace pruning: Using adaptive filter representations to improve training of sparse cnns." Proceedings of the IEEE/CVF conference on computer vision and pattern recognition. 2022
[3] Sidak Pal Singh, Dan Alistarh. "WoodFisher: Efficient Second-Order Approximation for Neural Network Compression". Part of Advances in Neural Information Processing Systems 33. 2020

## Second Question: About the Superiority of FLS with respect to other second-order methods, memory usage and clock time.

Our experiments show that FLS outperforms the other second-order pruning methods, even if this improvement is small in some experiments. However, as noted by one of our reviewers, previous second-order approaches saw smaller performance gains over each other, and FishLeg has other advantages that complement the performance improvement:

- Implementation flexibility: FishLeg directly amortize $F^{-1}v$ products in an easy-to-evaluate $Q(\lambda)v$ form. This approach enables both (i) easy application of FishLeg to different NN architectures and (ii) the development of different approximations for each type of NN layer to trade off scalability and accuracy.

- Faster wall-clock time and lower memory consumption: the dense layer and convolutional layer parametrization proposed in this work present a good balance of scalability and accuracy. Below, we provide a time comparison between FLS, oBERT and M-FAC as a function of number of samples collected. In these results, all algorithms implement the block (size 50) approximation for a fair comparison. Time is recorded during the re-computation of the Fisher inverse using the corresponding number of collected samples with batch size 16.


	|  Time Complexity \ Sample Size  | 512  | 2048      | 10k    | 50k     |
	|----------|--------|-------------|----------|------------|
	| FLS      | 0.096  |   0.239      |  0.981    | 4.755       |
	| oBERT    | 0.781  |   3.040      |  14.091    | 74.852       |
	| M-FAC     | 0.188  |   2.785      |  13.591    | 67.649       |

	Memory costs of the three algorithms are shown below during inverse-Fisher(Hessian)-vector-product computations as a function of the number of parameters.

	|  Memory Cost \ Network Size  | 500  | 40k      | 250k    |
	|----------|--------|-------------|----------|
	| FLS      | 911MB  |   913MB      |  937MB    |
	| oBERT    | 983MB  |   7107MB      |  OOM    |
	| M-FAC     | 981MB  |   1003MB      |  1033MB    |

	We will add these results to our supplementary materials.

---

### Meta-Review · Area_Chair_RJjo · 2023-12-06

**Metareview:**

In this paper, the authors use a novel formulation of the inverse Fisher information matrix to prune deep neural networks.  They then develop a block-wise approximation, reminiscent of the KFAC approximation, to achieve an efficient algorithm for doing the pruning.  In experiments on an autoencoder on MNIST and a RESNET-50 on Imagenet, the authors demonstrate better accuracy at a specified sparsity than baseline approaches (oBERT and M-FAC).  The reviews were overall mixed but leaning reject, with scores of 6, 5, 3, 3.  The reviewers found the approach well-motivated as making large deep networks more efficient is obviously valuable and interesting to the community.  The reviewers also found the approach technically sensible.  However, the reviewers didn't find the empirical evaluation of the method convincing, and some important citations/baselines were missing.  The major issue seems to be in the empirical evaluation.  The reviewers asked for experiments that demonstrate the efficiency of the method and, in particular, didn't find the scale of the problems presented particularly convincing.  While it's understandable that not all researchers have access to resources to train and evaluate the largest contemporary models, I agree that the presented problems aren't particularly motivating.  While LLM experiments aren't necessary, they could be helpful to motivate the work.  It's not hard to find reasonably tractable open-source models and a variety of interesting benchmark metrics.  This seems like an exciting start but, unfortunately, this paper doesn't seem quite ready for ICLR.  Strengthening the empirical evaluation would go a long way to improving the impact of the paper for a future submission.

**Justification For Why Not Higher Score:**

Too weak empirically.  The paper presents a method that only makes sense on large models, but they didn't evaluate on any even slightly large models.

**Justification For Why Not Lower Score:**

NA

---

### Decision · Program_Chairs · 2024-01-16

Reject